# Evaluating use of mass-media communication intervention 'MTV-Shuga' on increased awareness and demand for HIV and sexual health services by adolescent girls and young women in South Africa: an observational study

Natsayi Chimbindi [1,2] Nondumiso Mthiyane [1] Glory Chidumwa,[1,3] Thembelihle Zuma,[1,2] Jaco Dreyer,[1] Isolde Birdthistle [4] Sian Floyd [5] Nambusi Kyegombe,[6] Chris Grundy,[4] Cherie Cawood,[7] Siva Danaviah,[1] Theresa Smit,[1] Deenan Pillay,[8] Kathy Baisley,[1,9] Guy Harling [1,2] Janet Seeley [1,10] Maryam Shahmanesh[1,2]

NC and NM are joint first authors.

For numbered affiliations see end of article.

**Correspondence to**
Professor Maryam Shahmanesh;
m.shahmanesh@ucl.ac.uk

## ABSTRACT

**Objective** To investigate the effect of exposure to MTV Shuga:Down South' (MTVShuga-DS) during the scale-up of combination HIV-prevention interventions on awareness and uptake of sexual reproductive health (SRH) and HIV-prevention services by adolescent girls and young women (AGYW).

**Design** One longitudinal and three cross-sectional surveys of representative samples of AGYW.

**Setting** AGYW in four South African districts with high HIV prevalence (>10%) (May 2017 and September 2019).

**Participants** 6311 AGYW aged 12–24.

**Measures** Using logistic regression, we measured the relationship between exposure to MTV Shuga-DS and awareness of pre-exposure prophylaxis (PrEP), condom use at last sex, uptake of HIV-testing or contraception, and incident pregnancy or herpes simplex virus 2 (HSV-2) infection.

**Results** Within the rural cohort 2184 (85.5%) of eligible sampled individuals were enrolled, of whom 92.6% had at least one follow-up visit; the urban cross-sectional surveys enrolled 4127 (22.6%) of eligible sampled individuals. Self-report of watching at least one MTV Shuga-DS episode was 14.1% (cohort) and 35.8% (cross-section), while storyline recall was 5.5% (cohort) and 6.7% (cross-section). In the cohort, after adjustment (for HIV-prevention intervention-exposure, age, education, socioeconomic status), MTVShuga-DS exposure was associated with increased PrEP awareness (adjusted OR (aOR) 2.06, 95% CI 1.57 to 2.70), contraception uptake (aOR 2.08, 95% CI 1.45 to 2.98) and consistent condom use (aOR 1.84, 95% CI 1.24 to 2.93), but not with HIV testing (aOR 1.02, 95% CI 0.77 to 1.21) or acquiring HSV-2 (aOR 0.92, 95% CI 0.61 to 1.38). In the cross-sections, MTVShuga-DS was

associated with greater PrEP awareness (aOR 1.7, 95% CI 1.20 to 2.43), but no other outcome.

**Conclusions** Among both urban and rural AGYW in South Africa, MTVShuga-DS exposure was associated with increased PrEP awareness and improved demand for some HIV prevention and SRH technologies but not sexual health outcomes. However, exposure to MTVShuga-DS was low. Given these positive indications, supportive programming may be required to raise exposure and allow future evaluation of edu-drama impact in this setting.

---

**STRENGTHS AND LIMITATIONS OF THIS STUDY**

⇒ Evaluated the real-world reach of nationally broadcast edu-drama focusing on adolescent sexual health in South Africa.

⇒ Data collection focused on a vulnerable population of adolescent girls and young women (AGYW) across four diverse high HIV-burden districts of South Africa that included both rural and urban settings.

⇒ Strength of the study is measurement of exposure to MTV Shuga on HIV and sexual reproductive health outcomes in representative sample of AGYW during a real-world implementation in different South African settings.

⇒ The limitation of observational studies is that they do not infer the cause-and-effect relationship, in this case, we cannot ascertain causality/impact of exposure to MTV-Shuga on uptake of sexual health promotion and innovative technologies.

---

## INTRODUCTION

HIV remains one of the gravest health problems facing young people living in

sub-Saharan Africa. There are over 7.7 million people living with HIV in South Africa (SA), with more than 200 000 new HIV infections annually in those aged 15–49 years.[1] The highest incidence is in adolescent girls and young women (AGYW) (15–24 years).[1 2] Although HIV incidence has been declining in SA, a 43% decline in the overall incidence rate between 2012 and 2017, from 4.0 to 2.3 seroconversion events per 100 person-years among 15–49 years old; it still remains high among AGYW in SA.[3] In uMkhanyakude, HIV incidence was lower during roll-out of combination HIV prevention for AGYW (2016–2018) than in the previous 5-year period among females aged 15–19 years (4.5 new infections per 100 person-years as compared with 2.8; and lower among 20–24 years (7.1/100 person-years as compared with 5.8).[4]

In response to the high HIV incidence in young people, the South African government launched the 'She Conquers Campaign', and the US President's Emergency Fund for AIDS Relief and others are supporting the roll-out of Determined, Resilient, Empowered, AIDS free, Mentored and Safe (DREAMS).[5–7] These programmes provide an evidence-based combination HIV prevention package, including HIV testing and counselling for AGYW and their male sexual partners, alongside universal test and treat and improved sexual health services.[8 9]

However, the key ingredient to the success of these multicomponent interventions will be the extent to which AGYW and their male partners at most risk of HIV will uptake and adhere to the active components of the intervention. This is challenging: uptake and retention of contraception and HIV treatment cascade by young people, even within population-wide Universal Test and Treat trials, has been suboptimal.[2 10 11] Data from the baseline analysis for the DREAMS impact evaluation in uMkhanyakude district, rural KwaZulu-Natal (KZN) in 2015 suggest that less than 40% of girls (15–19 years) and boys and young men (15–29 years) had ever tested for HIV; linkage to HIV treatment was even lower.[12] Contraception use prevalence was 20% in girls (15–19 years) and 50% in young women (20–24 years) and 21% of girls aged 15–19 years had ever been pregnant.[2]

It is against this backdrop that the fifth series of Shuga: 'MTV Shuga:Down South' (MTV Shuga-DS), a mass-media serial edu-drama designed for SA, was broadcast on free-to-air SA national television (TV). From 8 March 2017, MTV Shuga aired one episode per week for 12 weeks (with repeats). MTV Shuga is a mass-media behaviour change campaign that aims to improve sexual and reproductive health rights (SRHR). At the centre of the campaign, which includes radio and social media, is a TV-drama that weaves messages about HIV, family planning, transactional and intergenerational sex, sexual identity, safer and healthy sexual relationships, into storylines with young characters (http://www.mtvshuga.com/show/series-5/MTV Shuga-down-south/).

Mass-media campaigns have the potential to reach a large number of people and have been shown to improve knowledge and health behaviour of a range of health conditions, with more recent data suggesting that theoretically informed and targeted interventions are more likely to have an effect.[13 14] MTV Shuga-DS was designed to reduce HIV-related risk behaviour and improve SRHR outcomes in adolescents and young adults in SA. This was expected to be achieved through increasing young people's awareness of their SRHR and demand for, and uptake of HIV and sexual reproductive health (SRH) prevention and treatment technologies. The show's characters explicitly model how to discuss issues that are sensitive or taboo. MTV Shuga use the technique of 'melodramas', where drama is created through the battles between stereotypical goodies and baddies, and the way in which the 'transitional' (often empathetic) character, begins as ambivalent but changes into a positive role model to promote positive behaviour change.[15] This is a deliberate method to immerse the audience in the action, rather than passively watching or listening.[16] AGYW, or at least early adopters, are anticipated to be immersed in the serial, able to classify and identify with the transitional characters and their outcomes. Pathways to behaviour change through MTV Shuga, therefore, relate to the extent to which the observer, including early adopters, are immersed and critically engaged with the story. It also depends on a context which is supportive rather than disruptive (see the conceptual framework figure 1).[17]

A cluster randomised controlled trial of community viewings of MTV Shuga in Nigeria found that exposure to MTV Shuga significantly improved HIV knowledge and testing in both sexes, the intervention arm showing 35% more likely to test for HIV than the control arm. There was also a 60% reduction in genital chlamydia as a marker of recent sexual risk in women, among those exposed to MTV Shuga compared with those who were not.[18] There were, however, fewer changes in social norms, particularly around gender-based violence. Further work suggested that the impact was greatest in those who were immersed in the narrative. The importance of immersion (classification of characters and identifying with them and observing outcomes) coupled with critical participation and an enabling context were also found in a thematic analysis looking at how storylines in MTV Shuga-DS shaped awareness, knowledge and opinions of sexual health and personal relationships among young people in SA.[17]

While the Nigerian and SA studies provide evidence for the efficacy of MTV Shuga impacting on SRHR and HIV-testing behaviours in exposed individuals, there is little evidence of how this will translate into a population-level effect when nationally broadcast and in less controlled environments. In particular, it is not clear how the impact will spill-over to non-viewers and how innovations will diffuse when shown and watched by adolescents and young adults in a real-world scale-up. It is also unclear how such impact will differ according to: setting (rural or urban); differential digital literacy and access to social media (geographically and socioeconomically), and the

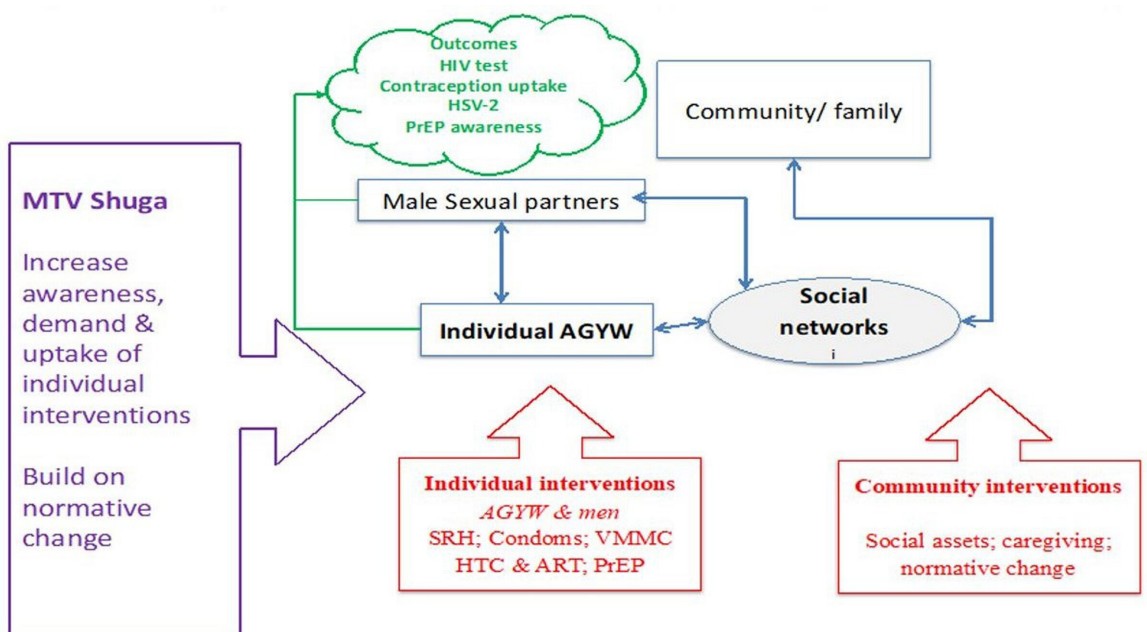

**Figure 1** Showing conceptual framework for MTV Shuga impact on HIV prevention on AGYW in uMkhanyakude. AGYW, adolescent girls and young women; ART, HIV antiretroviral therapy; HSV-2, herpes simplex virus 2; HTC, HIV Testing and Counselling; PrEP, pre-exposure prophylaxis; SRH, sexual reproductive health; VMMC, Voluntary Male Medical Circumcision.

dose and context of watching (shared viewing with family, friends or individual experience through social media).

We use the opportunity of MTV Shuga being broadcast in the context of an impact evaluation of DREAMS roll out (January 2016–September 2019) in a representative, population-based sample of young people[12] to describe the real-world reach of nationally broadcast MTV Shuga-DS. Further, we explore the hypothesis that exposure to a mass-media serial edu-drama, like MTV Shuga, will improve SRHR outcomes by increasing demand for, and uptake of, existing combination individual and community-based SRH and HIV prevention services for AGYW in four diverse settings including uMkhanyakude, a socioeconomically deprived rural district with an extremely high burden of HIV: 40% antenatal HIV prevalence and an annual HIV incidence of 5% in girls (15–19 years) and 8% in young women (20–24 years)[2 19] and in three high prevalence urban districts (HIV prevalence of greater than 10%) of City of Johannesburg, Ekurhuleni and eThekwini.

## METHODS
### Study design
We employed a longitudinal cohort and cross-sectional surveys of representative samples of AGYW aged 12–24 in four districts of SA with a high burden of HIV to measure the reach of MTV Shuga-DS. Data were collected between May 2017 and September 2019.

We used baseline and follow-up data from a nested cohort of 2184 AGYW aged 13–22 years, enrolled in 2017 for the DREAMS impact evaluation. The cohort is nested in a large population-based longitudinal HIV surveillance study, in the uMkhanyakude district of KZN.[20 21] A random sample of 3013 AGYW was selected from the surveillance population, stratified by age (13–17 years and 18–22 years) and geography, and invited to enrol in the nested cohort. Baseline interviews were conducted between May 2017 and February 2018 and follow-up interviews April 2018 and September 2019 in the local language (isiZulu) using a structured quantitative questionnaire programmed in REDCap software.[12]

The cross-sectional survey was conducted on a household-based representative sample of 4127 AGYW (between the ages 12 and 24 years) in 3 high prevalence (City of Johannesburg, Ekurhuleni and eThekwini) districts. Between August 2017 and July 2018, a stratified cluster-based sampling strategy was used to select 18 500 AGYW aged 12–24 eligible for a cross-sectional survey of individuals, based on an expected response rate of 80%.[22]

The interview included questions on sociodemographics, general health, exposure to DREAMS and to MTV Shuga, sexual relationships, awareness and uptake of DREAMS and DREAMS-like services, migration and gender norms across the four districts. A dried blood spot (DBS) was taken at baseline and follow-up for herpes simplex virus type 2 (HSV-2) antibody testing in the uMkhanyakude district. For sexual behaviour questions, participants were given a tablet computer to complete a self-interview; the fieldworker was available to provide support as needed.

### Study setting and population
The cohort was nested within the Africa Health Research Institute (AHRI) triannual demographic surveillance of a population of approximately 150 000 people who are

members of 15 000 geocoded households in an area of 845 km².[20] The study area is mostly rural and poor with high levels of youth unemployment (over 85% of those aged 18–24 are unemployed).[2 19]

The cross-sectional survey was conducted in three districts which were mostly urban with more towns and townships compared with AHRI surveillance area. The three study districts (City of Johannesburg, Ekurhuleni and eThekwini) consist of an estimated 12 073 421 individuals.

The eThekwini district in KZN province is among those with the highest HIV prevalence (16.8% HIV prevalence in 2016) in SA. Over two-thirds (68%) of eThekwini is considered rural and 32% urban. About 11 963 (3%) AGYW in eThekwini are estimated to be living with HIV. The Gauteng province (GP), while geographically the smallest, is the most populous province in SA. GP has the fifth highest provincial HIV prevalence in the country with a prevalence of 11.1% among those aged 15–49 years old in 2016. The HIV prevalence in the two districts, City of Johannesburg and Ekurhuleni, is 11.1% and 14.3% among 15–49 years, respectively. Both districts are densely populated and have high levels of industrialisation. The HIV prevalence among AGYW (15–24 years old) in the City of Johannesburg is 3%, and similarly 3% in Ekurhuleni in 2012.

### Variables and measurement
#### Outcomes definitions

We measured the effect of exposure to MTV Shuga on awareness and uptake of HIV prevention and SRHR outcomes at the follow-up visit. Our outcomes were: (1) self-reported HIV-testing in the past 12 months; (2) awareness of pre-exposure prophylaxis (PrEP) for HIV prevention; (3) condom use at last sex; (4) use of contraception; (5) any new pregnancy since baseline and (6) any new teenage pregnancy (restricted to those under the age of 20). AGYW were considered to use contraception if they self-reported using pill, injection, intrauterine device, implant, sterilisation or self-reported consistent condom use (using condoms as a contraceptive method and at last sex). Condomless sex was calculated using 2019 data among participants who reported having had sex with the most recent partner in the past 12 months. Recent pregnancy was calculated as any new pregnancy that occurred between baseline and 2019, while teenage pregnancy was calculated as any new pregnancy that occurred between baseline and 2019 among participants aged below 20 years. We also examined the effect of exposure to MTV Shuga on incident HSV-2 infections, among those who were HSV-2 negative at baseline.

#### Exposure definitions

Exposure to MTV Shuga was defined as ever watched MTV Shuga between 2017 and 2018. The level of exposure was measured based on the content of the series MTV Shuga, defined using 15 questions used to assess knowledge of content of MTV Shuga series. A composite score was developed summing-up the correct responses. The scores ranged between 2 and 14, and the median being 4. The median was used as a cut-off to define level of exposure among those who watched the series. Consequently, the level of exposure was categorised into three levels: high (watching an MTV Shuga and being able to correctly respond to five or more questions on content); medium (watching programme and being able to correctly respond to less than five questions) and none (not watched any MTV Shuga). This was further categorised into three levels: high (watching MTV Shuga and being able to recall the content from MTV Shuga); medium (watching programme, but unable to recall content) and none (not being aware of and not watched any MTV Shuga).

#### Potential confounding variables

We included sociodemographic and sexual behaviour characteristics of AGYW that were measured at baseline and exposure to HIV prevention. The sociodemographic variables included age (as measured at follow-up), household socioeconomic status (SES), education broken down by those who are still in school and those who have completed school, geographical area (rural or peri-urban/urban), and migration in the last 12 months. The SES variable was constructed using principal component analysis based on ownership of household assets and characteristics such as access to piped water, type of toilet, electricity and type of cooking fuel.[23] Further, potential individual-level confounders measured included exposure to DREAMS (defined either as ever been invited to participate in any of the DREAMS activities or ever used any of the DREAMS HIV prevention interventions in the past 12 months or since 2016) and phone ownership at baseline.

#### Laboratory

The HerpeSelect 2 ELISA IgG assay (FOCUS Diagnostics, Cypress, California, USA) for the qualitative detection of human IgG class antibodies to HSV-2 was used on DBS samples collected on Whatman 903 filter cards. The HerpeSelect 2 ELISA IgG assay uses purified type-specific gG-2 antigen immobilised on polystyrene microwells reducing the cross-reactivity issues as seen with viral lysate assays.[24] The assay is validated for use with serum samples but was optimised for use with DBS in the AHRI Diagnostic Research Laboratory following comparative testing with plasma samples. During the initial evaluation of the HerpeSelect 2 ELISA IgG a select number of plasma samples were also tested by an external accredited pathology laboratory. A 6 mm diameter punch of a DBS spot was incubated overnight in 150 µL assay diluent for no more than 16 hours at 4°C. The assay was performed with 50 µL of the eluent in accordance with the manufacturer's instructions. Following a disproportionately high number of positive results based on other studies and our experience, we multiplied the mean cut-off calibrator

absorbance values by a factor of 1.5 before determining the index value for each sample.[25 26] The HerpeSelect 2 ELISA IgG results are reported as positive (index value >1.10), equivocal (index value of ≥0.90 but ≤1.10) or negative (index value <0.90). All initial equivocal results were retested and those that retested equivocal are reported as equivocal. An incident HSV-2 was defined as having been negative at baseline and positive at follow-up. Those who were equivocal at either baseline or follow-up were not considered as a seroconversions.

### Statistical analyses

We conducted two separate analyses for cohort and cross-sectional data. For the nested cohort, we included only participants who had data available at baseline and follow-up. We used χ2 tests to compare baseline characteristics between AGYW who did and did not have any exposure to MTV Shuga. We used logistic regression to examine the effect of MTV Shuga on health outcomes, adjusting for exposure to DREAMS and all other potential confounders. Potential effect modification of MTV Shuga by exposure to DREAMS was examined by fitting an interaction term to fully adjusted model: likelihood ratio tests were used to compare models with and without interaction terms.

We calculated the proportion of AGYW who reported an outcome (condomless sex, recent pregnant) or tested positive for HSV-2 at 12-month or 24-month follow-up; and estimated associations between MTV Shuga and each outcome using a logistic regression, adjusting for potential confounders (age, household and individual sociodemographic characteristics and sexual behaviour). For HSV-2 incidence, we included participants who tested negative at baseline and had at least one follow-up test result. For DREAMS exposure, we included data collected at baseline and 12-month follow-up. For health outcomes (consistent condom use, modern contraception, HIV testing, PrEP awareness), we used data collected at 24-month follow-up; and for HSV-2 and pregnancy incidence we used data collected at 12-month and 24-month follow-up.

Propensity score logistic regression adjustment was used to estimate the causal effect of MTV Shuga on health outcomes. A propensity or probability of being exposed to MTV Shuga was measured by fitting a logistic regression with MTV Shuga exposure as an outcome and potential confounders. A logistic regression models adjusting for propensity scores were then used to predict the probability of an outcome for all participants and separately by age group, under two scenarios: (1) exposed to MTV Shuga and (2) not exposed to MTV Shuga. The predicted probabilities were then used to calculate the marginal risk difference, prevalence ratio and OR. CIs were generated by using a bootstrap procedure, repeating the estimation procedure described above in 1000 samples that were drawn with replacement from the complete dataset and calculating 95% CIs from the resulting bootstrap distribution using the 2.5% and 97.5% percentiles. We also used propensity score stratification and probability weighting methods to check the consistency of our findings.

For cross-sectional survey, we used χ2 test to compare the characteristics of AGYW who did and did not have any exposure to MTV Shuga; and logistic regression models adjusting for potential confounders were used to examine the association between exposure to MTV Shuga and health outcomes. Sampling weights were applied to achieve proportionality between groups of participants in the survey.

All analyses were performed using Stata V.15 (StataCorp).

### Reporting

The STROBE (STrengthening the Reporting of OBservational studies in Epidemiology) reporting guidelines were used to guide synthesis and standardise reporting of our results.[27]

### Patient and public involvement

The study did not involve patients. AHRI has a public engagement unit which conducts community engagement activities with the local communities as part of study findings dissemination. The community advisory board provided feedback of the study including design before approval from ethics review board. Study findings are being made publicly available to funders, participants and the public through webinars, study reports and open access journal articles.

## RESULTS

### Participants

Of 3013 potentially eligible AGYW randomly selected from the surveillance data set, 85.5% of those eligible consented to participate at baseline (figure 2). Of the 2184 eligible participants that were surveyed at baseline, 2016 (92.3%) had at least one follow-up visit and contributed data to this analysis. From the cross-sectional survey, 4127 (22.6%) eligible participants were surveyed.

### Awareness and exposure to MTV Shuga

MTV Shuga exposure at baseline was limited, with a total of only 308 (14.1%) respondents reported watching at least one episode. In the cross-sectional analysis of the three districts, a total of 1477 (35.8%) reported watching any MTV Shuga. In the nested cohort 121 (5.5%) recalled any storyline. Similarly in the cross-sectional snapshot 276 (6.7%) recalled the storyline (high exposure).

### Social demographic characteristics of adolescents and young people by exposure to MTV Shuga

Table 1 summarises the profiles of the nested cohort (n=2184) and cross-sectional surveys (n=4127) AGYW comparing those exposed to MTV Shuga with those not exposed. In summary those who had seen any MTV Shuga were more likely to be from households in the highest socioeconomic tertile (p<0.001) and more urbanised areas (p<0.001). They were also more likely to have

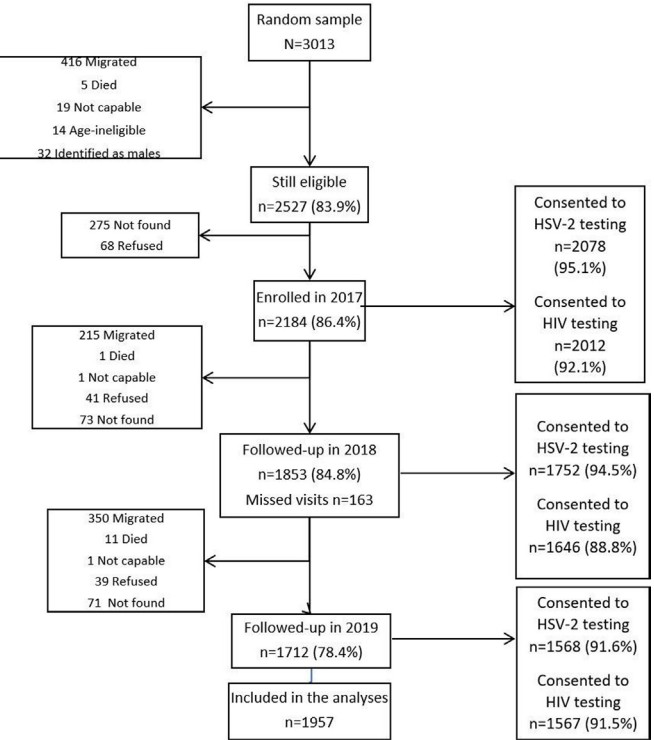

**Figure 2** Flow chart showing AGYW follow-up from nested cohorts 2017–2019. AGYW, adolescent girls and young women; HSV-2, herpes simplex virus 2.

also received DREAMS (p=0.015) (table 1). In the cross-sectional surveys (n=4127), MTV Shuga exposure was associated with older age (p<0.001), tertiary education (p<0.001) and never having sex (p<0.001) table 1.

After adjustment for confounders in nested cohorts (table 2), AGYW from wealthier households (adjusted OR (aOR) 2.04, 95% CI 1.27 to 3.30), peri-urban or urban areas (aOR 1.54, 95% CI 1.19 to 1.98) and those invited to DREAMS (aOR=1.48 95% CI 1.14 to 1.92) were more likely to be exposed to MTV Shuga than those from poor households, from rural areas and those not invited to DREAMS respectively. Similarly, after adjustment in the cross-sectional surveys (table 2), AGYW with higher education were more likely to be exposed to MTV Shuga (aOR 2.58, 95% CI 1.81 to 3.69) than those with less and those who ever had sex (aOR 0.68, 95% CI 0.57 to 0.82) were less likely to be exposed to MTV Shuga.

### Relationship between MTV Shuga exposure and HIV and SRHR outcomes

In the nested cohorts by 2019, overall 63.3% of those aged 14–23 knew their HIV status, 13.4% were consistently using contraception, 20.0% were using condoms consistently, and about one-third were aware of PrEP, 5% had a pregnancy and 15% acquired HSV-2. There were higher proportions of contraception use, condom use and PrEP awareness among those exposed to MTV Shuga (table 3). For survey sites (table 3), overall 85.0% knew their HIV status, over one-fifth, 22.6%, were using contraception and about half using condoms 48.4%. About 1/10th

(7.5%) were aware of PrEP, with higher proportions of these being among those exposed to MTV Shuga.

At follow-up, incident HSV-2 and teenage pregnancy were high, HSV-2 incidence was 15.26 and teenage pregnancy incidence was 9.86 per 100 person-years, respectively (table 4).

### MTV Shuga and HIV prevention and SRHR awareness and uptake

In the nested cohort after adjusting for age, education, SES, area and DREAMS, MTV Shuga exposure in the AGYW cohort was associated with significantly greater awareness of PrEP (aOR 2.06, 95% CI 1.57 to 2.70), contraception uptake (aOR 2.08, 95% CI 1.45 to 2.98) and consistent condom use (aOR 1.84, 95% CI 1.24 to 2.93). Watching MTV-Shuga was not associated with HIV testing (aOR 1.02, 95% CI 0.77 to 1.21) (figure 3 and online supplemental tables S1–S4). There was no effect modification by DREAMS exposure.

Similarly, in the cross-sectional analysis, after adjusting for age, education, district, migration and sexual history, exposure to MTV-Shuga watching was associated with greater awareness of PrEP (aOR 1.70, 95% CI 1.20 to 2.43). However, there was no association with contraception, lower self-reported HIV testing or condom use as shown in figure 4 (online supplemental table S5).

### Causal effect of MTV Shuga on health outcomes: HSV-2

The causal analysis similarly found no effect of MTV Shuga on HSV-2, with a risk difference of 1.10 95% CI (−2.82% to 5.38)%. Findings in the younger age group (aged 13–17) and the older age group[18–22] were similar (online supplemental table S6).

### DISCUSSION

In our study of the population-level effect of a national broadcast of a TV-based edu-drama on HIV prevention and SRHR, exposure to MTV Shuga was associated with higher awareness of a novel HIV prevention intervention (PrEP). However, despite a very high incidence of HSV-2 and teenage pregnancy, MTV Shuga exposure was not significantly associated with safer sexual behaviour, uptake of contraception and HIV-testing or prevention of teenage pregnancy. Notably though, the size of the relationship and direction of effect between exposure to MTV Shuga, condom use and markers of unprotected sex (HSV-2 and pregnancy) was consistent with a possible relationship. These findings may be partly explained by our finding that less than 1 in 12 of the target age group had any exposure to MTV Shuga and only half of these had high exposure (defined as watching the MTV Shuga South African series and being able to recall the content), suggesting that one of the limiting factors for the effective use of TV-based edu-drama maybe the dose that young people are exposed to, particularly in rural and resource-constrained settings most affected by HIV.

**Table 1** Baseline sociodemographic characteristics of adolescent girls and young women (13–22) in the nested cohort (n=2184) and (12–24) in the cross-sectional survey (n=4127) by exposure to MTV Shuga

| | Overall | | Exposed | | Not exposed | | |
|---|---|---|---|---|---|---|---|
| | n/N | % | n/N | % | n/N | % | P value |
| Baseline sociodemographic characteristics of adolescent girls and young women (13–22) by exposure to MTV Shuga in the nested cohort (n=2184) | | | | | | | |
| Age group (4 cats), 2017 | | | | | | | |
| 13–14 | 460/2184 | 21.1 | 72/308 | 23.4 | 388/1876 | 20.7 | |
| 15–17 | 688/2184 | 31.5 | 107/308 | 34.7 | 581/1876 | 31 | |
| 18–19 | 475/2184 | 21.7 | 60/308 | 19.5 | 415/1876 | 22.1 | |
| 20–22 | 561/2184 | 25.7 | 69/308 | 22.4 | 492/1876 | 26.2 | 0.216 |
| Currently in school | | | | | | | |
| No | 540/2184 | 24.7 | 67/308 | 21.8 | 473/1876 | 25.2 | |
| Yes | 1644/2184 | 75.3 | 241/308 | 78.2 | 1403/1876 | 74.8 | 0.192 |
| Socioeconomic status, 2018 | | | | | | | |
| Low | 255/2118 | 12 | 22/303 | 7.3 | 233/1815 | 12.8 | |
| Middle | 920/2118 | 43.4 | 110/303 | 36.3 | 810/1815 | 44.6 | |
| High | 943/2118 | 44.5 | 171/303 | 56.4 | 772/1815 | 42.5 | <0.001 |
| Urban or rural | | | | | | | |
| Rural | 1388/2165 | 64.1 | 165/305 | 54.1 | 1223/1860 | 65.8 | |
| Periurban/urban | 777/2165 | 35.9 | 140/305 | 45.9 | 637/1860 | 34.2 | <0.001 |
| Invited or received DREAMS, 2017/2018 | 1101/2184 | 50.4 | 175/308 | 56.8 | 926/1876 | 49.4 | 0.015 |
| Away from home in the last 12 months | 314/1853 | 17.0 | 45/228 | 15.6 | 269/1565 | 17.2 | 0.516 |
| Baseline sociodemographic characteristics of adolescent girls and young women (12–24) by exposure to MTV Shuga in the cross-sectional survey (n=4127) | | | | | | | |
| Age group, (N=4127) | | | | | | | |
| 12–14 | 958/4127 | 23.2 | 307/1477 | 20.1 | 651/2650 | 24.6 | |
| 15–19 | 1628/4127 | 39.5 | 599/1477 | 40.6 | 1029/2650 | 38.8 | |
| 20–24 | 1541/4127 | 37.3 | 571/1477 | 38.7 | 970/2650 | 36.6 | 0.022 |
| District | | | | | | | |
| City of Johannesburg | 1146/4127 | 27.8 | 476/1477 | 32.2 | 670/2650 | 25.3 | |
| Ekurhuleni | 1635/4127 | 39.6 | 521/1477 | 35.3 | 1114/2650 | 42.0 | |
| eThekwini | 1342/4127 | 32.5 | 480/1477 | 32.5 | 862/2650 | 32.5 | <0.001 |
| Highest education (N=4108) | | | | | | | |
| No schooling | 175/4108 | 4.3 | 99/1465 | 6.8 | 76/2646 | 2.9 | |
| Grades R–7 | 502/4108 | 12.2 | 142/1465 | 9.7 | 360/2646 | 13.6 | |
| Grades 8–12 | 2978/4108 | 72.5 | 1022/1465 | 69.8 | 1956/2646 | 74.0 | |
| Tertiary studies (complete/incomplete) | 453/4108 | 11.0 | 202/1465 | 13.8 | 251/2646 | 9.5 | <0.001 |
| Ever had sex with a boy/man (n=4108) | 1860/4108 | 45.3 | 621/1469 | 42.3 | 1239/2639 | 47.0 | 0.004 |
| Away from home in the last 12 months (n=4121) | 183/4121 | 4.4 | 66/1474 | 4.5 | 117/2647 | 4.4 | 0.932 |

DREAMS, Determined, Resilient, Empowered, AIDS free, Mentored and Safe.

Our inability to find a measurable population effect of mass-media edu-drama behaviour change campaign compared with the trial findings of the randomised controlled trial is disappointing.[18] However, first the direction and size of the relationships suggest that we may have been able to see an effect if the proportion exposed had been greater than 7%. Second, AGYW in urban settings were more likely to have been exposed to MTV Shuga and they are also more likely to be living in small towns and townships. Data from our settings suggest that young people in small towns and townships are more vulnerable to HIV[28] and sexual risk,[29] and therefore, it is possible that a real effect of MTV Shuga on this group was masked by their greater risk for the outcome. Due to the low numbers with high levels of exposure to MTV Shuga, we did not have the power to

**Table 2** Factors associated with exposure to MTV Shuga in the nested cohort of AGYW aged 13–22 (n=2184) and in the cross-sectional analysis of AGYW aged 12–24 (n=4127)

| | Unadjusted | | Adjusted—all | | |
|---|---|---|---|---|---|
| | OR | 95% CI | OR | 95% CI | P value |
| **Factors associated with exposure to MTV Shuga in the nested cohort of AGYW aged 13–22 (n=2184)** | | | | | |
| **Age group, 2017** | | | | | |
| 13–14 | 1 | | 1 | | |
| 15–17 | 0.99 | 0.72 to 1.37 | 0.93 | 0.66 to 1.29 | |
| 18–19 | 0.78 | 0.54 to 1.13 | 0.77 | 0.51 to 1.14 | |
| 20–22 | 0.76 | 0.53 to 1.08 | 0.78 | 0.50 to 1.21 | 0.555 |
| **Currently in school** | | | | | |
| No | 1 | | 1 | | |
| Yes | 1.21 | 0.91 to 1.62 | 1.06 | 0.78 to 1.43 | 0.726 |
| **Socioeconomic status, 2018** | | | | | |
| Low | 1 | | 1 | | |
| Middle | 1.44 | 0.89 to 2.33 | 1.27 | 0.78 to 2.07 | |
| High | 2.35 | 1.47 to 3.74 | 2.04 | 1.27 to 3.30 | <0.001 |
| **Site** | | | | | |
| Rural | 1 | | 1 | | |
| Periurban/urban | 1.63 | 1.28 to 2.08 | 1.54 | 1.19 to 1.98 | 0.001 |
| **Invited or received DREAMS, 2017/2018** | | | | | |
| No | 1 | | 1 | | |
| Yes | 1.35 | 1.06 to 1.72 | 1.48 | 1.14 to 1.92 | 0.003 |
| **Factors associated with exposure to MTV Shuga in the cross-sectional analysis of AGYW aged 12–24 (n=4127)** | | | | | |
| **District** | | | | | |
| City of Johannesburg | 1 | | 1 | | |
| Ekurhuleni | 0.69 | 0.54 to 0.89 | 0.66 | 0.52 to 0.84 | 0.001 |
| eThekwini | 0.79 | 0.60 to 1.04 | 0.78 | 0.59 to 1.04 | 0.087 |
| **Age group** | | | | | |
| 12–14 | 1 | | 1 | | |
| 15–19 | 1.27 | 1.06 to 1.53 | 1.18 | 0.95 to 1.46 | 0.126 |
| 20–24 | 1.22 | 1.01 to 1.48 | 1.16 | 0.90 to 1.49 | 0.256 |
| **Highest education level** | | | | | |
| Grades R–7 | 1 | | 1 | | |
| No schooling | 2.88 | 1.81 to 4.58 | 3.29 | 2.01 to 5.38 | <0.001 |
| Grades 8–12 | 1.38 | 1.09 to 1.75 | 1.42 | 1.08 to 1.86 | 0.011 |
| Complete or incomplete tertiary | 2.29 | 1.70 to 3.09 | 2.58 | 1.81 to 3.69 | <0.001 |
| **Ever had sex with a boy/man** | | | | | |
| No | 1 | | 1 | | |
| Yes | 0.82 | 0.71 to 0.95 | 0.68 | 0.57 to 0.82 | <0.001 |
| **Away from home in last 12 months** | | | | | |
| No | 1 | | 1 | | |
| Yes | 1.15 | 0.80 to 1.64 | 1.21 | 0.83 to 1.77 | 0.309 |

AGYW, adolescent girls and young women; DREAMS, Determined, Resilient, Empowered, AIDS free, Mentored and Safe.

explore this hypothesis by looking at effect modification by geography or SES.

The finding of a differential association of MTV Shuga exposure with awareness of PrEP, compared with uptake of HIV and contraception, suggests that while educational mass entertainment may be able to increase awareness and possibly demand for a service, it does not impact on accessibility of the service, that is, it impacts

**Table 3** HIV and SRHR outcomes by exposure to MTV Shuga nested cohort 13–22 years old (n=2167) and HIV and SRHR outcomes by exposure to MTV Shuga weighted for sampling cross sectional survey 12–24 years old (n=4127)

| | Overall | | Exposed (n=308) | | Not exposed (n=1878) | | |
|---|---|---|---|---|---|---|---|
| | n/N | % | n/N | % | n/N | % | P value |
| HIV and SRHR outcomes by exposure to MTV Shuga nested cohort 13–22 years old (n=2167) | | | | | | | |
| Knowledge of HIV status, 2019 | 1083/1712 | 63.3 | 175/283 | 61.8 | 908/1429 | 63.5 | 0.587 |
| Modern contraception, 2019 | 221/1651 | 13.4 | 56/271 | 20.7 | 165/1380 | 12 | <0.001 |
| Consistent condom use, 2019 | 168/838 | 20 | 41/141 | 29.1 | 127/697 | 18.2 | 0.003 |
| Aware of PrEP, 2019 | 523/1712 | 30.5 | 124/283 | 43.8 | 399/1429 | 27.9 | 0.302 |
| Pregnant in 2018/2019 | 124/2184 | 5.7 | 20/308 | 6.5 | 104/1876 | 5.5 | 0.504 |
| Teenage pregnancy | 72/1395 | 5.16 | 55/1187 | 4.63 | 17/208 | 8.17 | 0.033 |
| HSV-2 2018/2019 | 241/1562 | 15.4 | 35/237 | 14.8 | 206/1325 | 15.5 | 0.760 |
| HIV and SRHR outcomes by exposure to MTV Shuga weighted for sampling cross sectional survey 12–24 years old (n=4127) | | | | | | | |
| PrEP awareness (N=4127) | 310/4127 | 7.5 | 148/1477 | 10.0 | 162/2650 | 6.1 | <0.001 |
| HIV test (self-report) (N=2529) | 2156/2529 | 85.3 | 656/797 | 82.3 | 1500/1732 | 86.6 | 0.005 |
| Condom use at last sex (N=1898)* | 918/1898 | 48.4 | 320/640 | 50.0 | 598/1258 | 47.5 | 0.310 |
| Contraception use (N=4127) | 934/4127 | 22.6 | 302/1477 | 20.5 | 632/2650 | 23.9 | 0.012 |

*Restricted to those who ever had sex with man.
HSV-2, herpes simplex virus 2; PrEP, pre-exposure prophylaxis; SRHR, sexual and reproductive health rights.

the first two steps of the prevention cascade and not the final step.[30] Well-described barriers to uptake of HIV testing and contraception in this area are internalised and externalised stigma, fear of judgement from healthcare workers and the social costs of accessing care in busy primary healthcare settings.[31–33] Behaviour change intervention including mass communication campaigns can be constrained or facilitated by the context in which people live.[17 34] To optimise MTV Shuga's effect, there may need to be parallel innovations in SRHR and HIV service delivery that makes the services easier to access. We aim to test this hypothesis by providing community-based delivery of HIV and SRHR services in the context of the MTV Shuga.

The behaviour change theory that underpins edu-drama as a vehicle for mass behaviour change communication[16] explicitly suggests that the audience, and especially the early adopters, need to be actively watching, rather than passively watching or listening.[17 18] TV watching in rural homesteads can be in the context of large, often grandparent-led households and competing chores and priorities. This coupled with the relatively late timing of the shows may explain why so few girls and young women were sufficiently engaged or immersed to be able to recall characters or story lines. Moreover, the timing of this analysis may have allowed insufficient time for early adopters to convey the message of the show.

**Table 4** Rate of SRHR outcome by exposure to MTV Shuga (nested cohort 13–22 years old n=2167)

| | Person-time | n with HSV-2 | Rate/100 person-years | 95% CI |
|---|---|---|---|---|
| HSV-2 incidence rate | | | | |
| MTV Shuga not exposed | 1303.0 | 206 | 15.81 | 13.79 to 18.12 |
| MTV Shuga exposed | 276.3 | 35 | 12.67 | 9.09 to 17.64 |
| Total | 1579.3 | 241 | 15.26 | 13.45 to 17.31 |
| Pregnancy incidence rates among all AGYW | | | | |
| MTV Shuga not exposed | 1068.7 | 104 | 9.73 | 8.03 to 11.79 |
| MTV Shuga exposed | 188.5 | 20 | 10.61 | 6.84 to 16.44 |
| Total | 1257.2 | 124 | 9.86 | 8.27 to 11.76 |
| Pregnancy incidence rates among girls below 19 years | | | | |
| MTV Shuga not exposed | 668.7 | 55 | 8.22 | 6.31 |
| MTV Shuga exposed | 123.9 | 17 | 13.72 | 8.53 |
| Total | 792.6 | 72 | 9.08 | 7.21 |

AGYW, adolescent girls and young women; HSV-2, herpes simplex virus 2; SRHR, sexual and reproductive health rights.

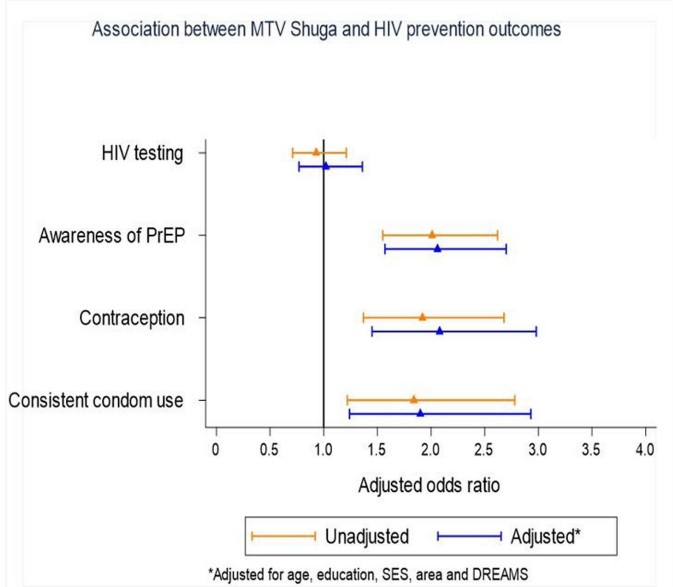
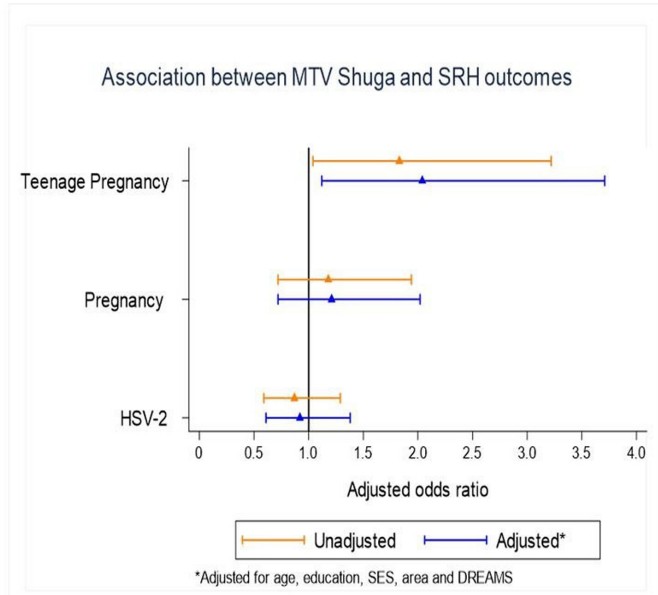

**Figure 3** Forest plots showing the association between MTV Shuga exposure and HIV prevention and SRHR awareness and uptake in the nested cohorts. DREAMS, Determined, Resilient, Empowered, AIDS free, Mentored and Safe; PrEP, pre-exposure prophylaxis; SES, socioeconomic status; SRH, sexual reproductive health; SRHR, sexual and reproductive health rights.

### Limitations

The limitations of our study are that a meaningful exposure to MTV Shuga was low and so while the size of the relationship and direction of effect between exposure to MTV Shuga, condom use and markers of unprotected sex (HSV-2 and pregnancy) suggested a possible effect, but we did not have the power to show a significant relationship between exposure to MTV Shuga and SRHR outcomes. We also do not have the power to see a difference by dose and immersion. Furthermore, as this is an observational study, we cannot exclude the possibility that those who are exposed to MTV Shuga are systematically different in ways that impact on the outcome of interest, for example, more exposed to social media and sexual health promotion and innovative technologies to support sexual health than those who were not.

### Conclusions and implications for the future

► Meaningful exposure to MTV Shuga was low across rural and urban settings in SA and so additional

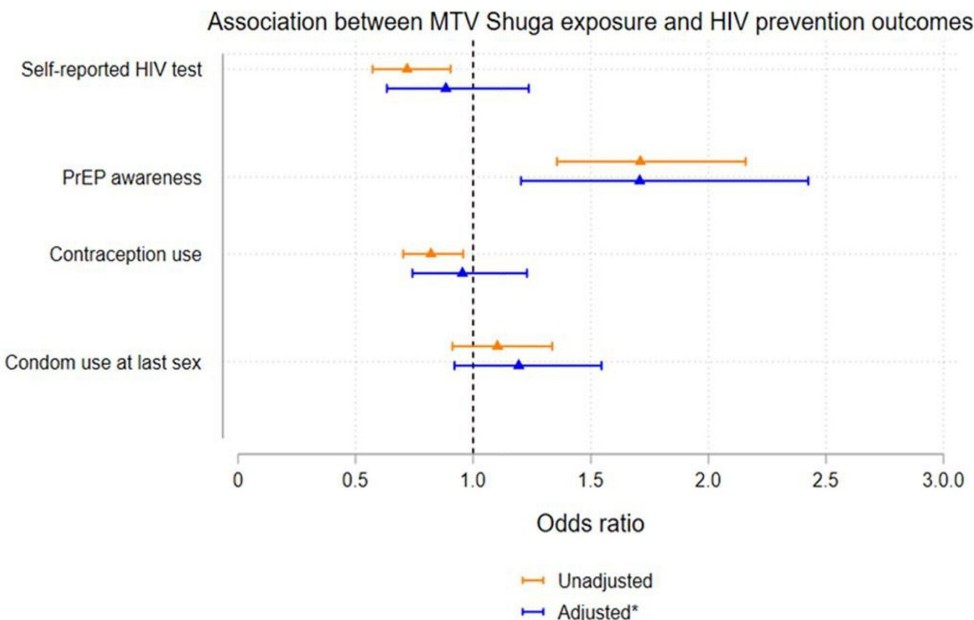

**Figure 4** Forest plots showing association between MTV Shuga exposure and HIV prevention and SRHR awareness and uptake in the cross-sectional surveys. PrEP, pre-exposure prophylaxis; SRHR, sexual and reproductive health rights.

efforts need to be made to reach young people and increase their immersion in promising edu-drama if it is to have the desired effect, especially in rural and deprived settings.

► MTV Shuga was an effective vehicle to raise awareness and promote newer HIV prevention technologies such as HIV PrEP.

► There was some suggestion that MTV Shuga improved uptake of some HIV prevention and sexual health technologies (contraception and condoms).

► There was less evidence from this observational study that it improved SRHR and HIV outcomes.

We highlight the importance of evaluating the real-world scale up of promising interventions to understand both the reach and population effect as well as inform interventions to increase impact and equity.

Efforts to increase exposure, which have been rolled out as part of MTV Shuga in SA, such as social media, school-based or community-based MTV Shuga film clubs will need to be evaluated, both to understand whether or not they increase exposure and coverage and improve SRH and HIV outcomes. However, to have a significant impact on the HIV and SRH prevention and treatment cascades, demand generation in AGYW needs to be delivered in parallel with accessible service delivery models that support adherence and retention.[30]

**Author affiliations**
[1]Africa Health Research Institute, KwaZulu-Natal, South Africa
[2]Institute for Global Health, University College London, London, UK
[3]Faculty of Health Sciences, University of the Witwatersrand, Wits Reproductive Health and HIV Institute, Johannesburg, South Africa
[4]Department of Population Health, London School of Hygiene and Tropical Medicine Faculty of Epidemiology and Population Health, London, UK
[5]Infectious Disease Epidemiology, London School of Hygiene & Tropical Medicine, London, UK
[6]MRC Uganda Virus Research Institute, London School of Hygiene & Tropical Medicine, London, UK
[7]Epicentre Health Research, Durban, South Africa
[8]Infection and Immunity, University College London, London, UK
[9]Faculty of Epidemiology and Population Health, London School of Hygiene and Tropical Medicine, London, UK
[10]Department of Global Health & Development, London School of Hygiene and Tropical Medicine, London, UK

**Acknowledgements** The authors are grateful to the communities of the Hlabisa subdistrict, Ekurhuleni district, Thekwini district and City of Johannesburg who contributed their data to this study and to all the staff at Africa Health Research Institute and EpiCentre who collected the data. We acknowledge Eskindir Shambullo for his input in this manuscript.

**Contributors** MS, NC, JD, TZ, NK, GH and CC developed the study tools and performed the research. MS, JS, DP, KB, IB and SF designed the research study. TS and SD conducted the laboratory analysis. NM supported by KB conducted the statistical analysis and GC, MS wrote the first and final draft of the paper with input from NC, NM, GH, JS, IB, GC, KB, CC, TS, TZ, DP and NK. All the authors critically reviewed the manuscript. All authors have approved the final draft of the paper. MS is the guarantor and accepts full responsibility for the work and the conduct of the study, has access to the data and controlled the decision to publish.

**Funding** The impact evaluation of DREAMS and MTV Shuga is funded by the Bill & Melinda Gates Foundation (OPP1136774 and OPP1171600, http://www.gatesfoundation.org). Foundation staff advised the study team, but did not substantively affect the study design, instruments, interpretation of data or decision to publish. The research leading to these results has received funding from the People Programme (Marie Curie Actions) of the European Union's seventh Framework Programme FP7/2007-2013 under REA grant agreement no 612216. MS National Institutes of Health 5R01MH114560-03 funding acknowledgement. This research was funded in whole, or in part, by the Africa Health Research Institute through the Wellcome (Strategic Core award: 201433/Z/16/A). For the purpose of open access, the author has applied a CC BY public copyright licence to any Author Accepted Manuscript version arising from this submission. NC is supported by a training fellowship from the National Institute for Health Research (NIHR) (using the UK's Official Development Assistance (ODA) Funding) and Wellcome (grant reference number 224309/Z/21/Z) under the NIHR-Wellcome Partnership for Global Health Research. GH is supported by a fellowship from the Royal Society and the Wellcome Trust (Grant number 210479/Z/18/Z). MS work is also supported by a BMGF 3ie grant at AHRI. This research was funded in whole, or in part, by the Wellcome Trust (grant number 210479/Z/18/Z). For the purpose of open access, the author has applied a CC BY public copyright licence to any author accepted manuscript version arising from this submission.

**Patient and public involvement** Patients and/or the public were involved in the design, or conduct, or reporting, or dissemination plans of this research. Refer to the Methods section for further details.

**Patient consent for publication** Not applicable.

**Ethics approval** The DREAMS Partnership impact evaluation protocol was reviewed and approved by the University of KwaZulu-Natal (UKZN) Biomedical Research Ethics Committee (BREC) (BFC 339/16), the London School of Hygiene & Tropical Medicine Research Ethics Committee (Ref 11835) and the AHRI Somkhele Community Advisory Board, the Associate Director of Science of the Center for Global Health (CGH) at the US Centers for Disease Control and Prevention (CDC) in Atlanta and the Department of Health, Province of KZN and Gauteng. Approval for the demographic surveillance and data collection in the clinics was granted by UKZN BREC. Participation in the study was voluntary. All participants provided separate informed consent to take part in data collection through the questionnaires and the HSV2 sero-survey. Consent for follow-up interviews was provided separately. For participants aged <18 years, written parental consent and participant assent were provided. All data were anonymised for analysis.

**Data availability statement** Data are available on reasonable request. Data are available from the Africa Health Research Institute data repository.

**ORCID iDs**
Natsayi Chimbindi http://orcid.org/0000-0003-3125-978X
Nondumiso Mthiyane http://orcid.org/0000-0002-9881-7267
Isolde Birdthistle http://orcid.org/0000-0001-5742-6588
Sian Floyd http://orcid.org/0000-0002-8615-7601
Guy Harling http://orcid.org/0000-0001-6604-491X
Janet Seeley http://orcid.org/0000-0002-0583-5272

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
