## [Reviewer comments · BMJ Open]

ARTICLE DETAILS

TITLE (PROVISIONAL)	Evaluating use of mass-media communication intervention 'MTV-Shuga' on increased awareness and demand for HIV and sexual health services by adolescent girls and young women in South Africa: An observational study
AUTHORS	Chimbindi, Natsayi; Mthiyane, Nondumiso; Chidumwa, Glory; Zuma, Thembelihle; Dreyer, Jaco; Birdthistle, Isolde; Floyd, Sian; Kyegombe, Nambusi; Grundy, Chris; Cawood, Cherie; Danaviah, Siva; Smit, Theresa; Pillay, Deenan; Baisley, Kathy; Harling, Guy; Seeley, Janet; Shahmanesh, Maryam

VERSION 1 – REVIEW

REVIEWER	Ngere, Sarah Hawi Maseno University, Public Health and Community Development
REVIEW RETURNED	30-Apr-2022

GENERAL COMMENTS	Manuscript title: 'MTV Shuga': Mass media communication in adolescent girls and young women in South Africa: Can it increase awareness and demand for HIV and sexual health technologies Journal: BMJ OPEN Manuscript ID: bmjopen-2022-062804 Corresponding author: Natsayi Chimbindi Reviewer: Sarah Hawi Ngere, BA, MPH Review summary According to my understanding the study was set to investigate the effect of exposure to MTV Shuga Down South (DS) during the scale-up of combination HIV-prevention intervention interventions (DREAMS) on awareness and uptake of SRH and HI prevention services by adolescent girls and young women (AGYW). The data was collected through a defined period of time. The researchers have concrete understanding of the topic area. I have only minor comments the manuscript is well written with clarity on all aspects. Section by section comments Page 2 Abstract
--

	Line 11: Under subsection, objectives, DREAMS acronym appears for the first time but authors do not give the meaning of what acronym stands for. Comment: No further comments. This section is well written. Page 4-6 Introduction The authors did a good job in this section, introducing the readers to the background of the problem and systematically stating out the problem and what gaps the study intends to fill Methods General comment/observation: I find this section well written. The design has rigor and reliability. Page 10 line 52-60 and page 11 line 1-13: Ethics, Ethical consideration for data management is missing. Suggestion: Consider including a sentence to describe how the data collection and storage were ethically sound. Results General comment/observation: Other than one comment stated below the section is clearly written. Page 12 line 11: Awareness and exposure to MTV Shuga, “MTV shuga exposure at baseline was low” Since exposure is not clearly defined on page 8 line 34 -52 I found it difficult to understand “low exposure” as stated in the line above because of how it was described therefore I think it should be either exposed or not exposed to MTV Shuga (please note that I understood there was a difference between exposure and level of exposure). Page 17 Discussion General comment/observation: The discussion section is well written with all major findings adequately discussed. Page 18 Limitations General comment/observation: The limitation section indicates enough evidence to guide future studies on the gaps that the study was unable to fill. I find the section well written. Line 51 “...effect we did not have the power to show significant...” this sentence may be missing “but” therefore the statement can be state as: “...effect but we did not have the power to show significant...” Conclusion and implications for the future
--	--

	General comment/observation: No comments for this section.
--	--

REVIEWER	Eaton, Andrew University of Regina, Faculty of Social Work - Saskatoon Campus
REVIEW RETURNED	09-Jun-2022

GENERAL COMMENTS	This is a compelling study of the outcomes of an edu-drama on adolescent girls and young women in South Africa. The authors conducted complex and multifaceted data collection, and I appreciated a good degree of transparency and clarity in the reporting. I have the following suggestions. Introduction  1. Line 11 - please state incidence of HIV infection amongst AGYW in South Africa 2. Paragraph 2 - Implied assumption that all AGYW are heterosexual and sexually active is problematic 3. It is jarring to see words such as 'good' and 'evil' in the context of highly stigmatised conditions such as HIV. I recommend providing examples of these characters and if 'evil' behaviour includes potentially risky sexual behaviour, that this be discussed and critiqued. 4. Last paragraph, line 13 - correct "young people" to "AGYW". Methods  1. Why are the sample sizes an approximation? Please provide the exact samples if possible. 2. Study design paragraph 3 - please cite where the cross-sectional survey data has already been published and/or the protocol, as you did with the nested cohort in the preceding paragraph. 3. Outcomes definitions lines 14-19 - This is a confusing way to state contraception methods included in the study. I recommend providing a comprehensive list of the actual methods instead. 4. Exposure definitions: Did you assess the number of MTV Shuga episodes watched? Results  1. With over 85% of participants having not watched MTV Shuga, the exposed and not exposed groups seem unweighted to a potentially problematic degree but I don't see detail of mitigating strategies.
---

REVIEWER	Li, Xianhong Central South University, Xiangya School of Nursing
REVIEW RETURNED	18-Jun-2022

GENERAL COMMENTS	This study evaluate a mass-media education program in the real world settings, and the results are meaningful and have potential for further intervention designs. This manuscript is well-written. However, since the study desgin is complicated, with cohort data and cross-sectional data, the author might draw a design flow-chart to elaborate how the data are from. For example, how many data collection in the cohort? When to collect the three cross-sectional surveys data (at three time piont)? how to analysis it? It seems the manucript did not clarify this, but just use logistic regression analysis (by using which surveys?).
---

	Besides, what the rational to both using cohort data and the cross-sectional data? Please make it clear.
--	--

VERSION 1 – AUTHOR RESPONSE

Reviewer: 1

Dr. Sarah Hawi Ngere, Maseno University

Comments to the Author:

I found this work exhilarating and well written. I have given a few minor comments for your consideration.

Response

Thank you for the feedback

Reviewer: 2

Dr. Andrew Eaton, University of Regina

Comments to the Author:

This is a compelling study of the outcomes of an edu-drama on adolescent girls and young women in South Africa. The authors conducted complex and multifaceted data collection, and I appreciated a good degree of transparency and clarity in the reporting. I have the following suggestions.

Response

Thank you for the comment and the useful suggestions, we have taken them into consideration as outlined below.

Introduction

1. Line 11 - please state incidence of HIV infection amongst AGYW in South Africa

Response 1

Thank you for raising this. We have now included AGYW HIV incidence in SA and more specifically from our study site, as follows:

“Although HIV incidence has been declining in South Africa, a 43% decline in the overall incidence rate between 2012 and 2017, from 4.0 to 2.3 seroconversion events per 100 person-years among 15-49 year old; it still remains high among AGYW in South Africa (3). In uMkhanyakude, HIV incidence was lower during roll-out of combination HIV prevention for AGYW (2016 to 2018) than in the previous 5-year period among 15- to 19-year-old females (4.5 new infections per 100 person-years as compared with 2.8; and lower among 20- to 24-year-olds (7.1/100 person-years as compared with 5.80.(4)”

2. Paragraph 2 - Implied assumption that all AGYW are heterosexual and sexually active is problematic

Response 2

We agree with the point the reviewer raises. However, this comes from the way the DREAMS programme was designed, focus was on sexually acquired HIV in AGYW and their male partners. We have now added the missing word 'their' to the sentence to bring out the aspect that DREAMS was looking at HIV risk among AGYW and their male sexual partners.

“These programmes provide an evidence-based combination HIV prevention package, including HIV-testing and counselling for adolescent girls and young women (AGYW) and their male sexual partners, alongside universal test and treat and improved sexual health services”

3. It is jarring to see words such as 'good' and 'evil' in the context of highly stigmatised conditions such

as HIV. I recommend providing examples of these characters and if 'evil' behaviour includes potentially risky sexual behaviour, that this be discussed and critiqued.

Response 3

Thank you for raising this issue and it is something that we explored in our qualitative data, where we found young people had a very nuanced understanding of the context in which they navigate sex (15 Kyegombe N, Zuma T, Hlongwane S, et al. A qualitative exploration of the salience of MTV-Shuga, an edutainment programme, and adolescents' engagement with sexual and reproductive health information in rural KwaZulu-Natal, South Africa. *Sexual and Reproductive Health Matters* 2022; 30(1): 2083809). We have edited the language here to clarify and added a reference to our qual paper for more detail as below.

Mass media campaigns, have the potential to reach a large number of people and have been shown to improve knowledge and health behaviour of a range of health conditions, with more recent data suggesting that theoretically informed and targeted interventions are more likely to have an effect (11, 12). MTV Shuga-DS was designed to reduce HIV-related risk behaviour and improve SRHR outcomes in adolescents and young adults in SA. This was expected to be achieved through increasing young people's awareness of their sexual and reproductive health rights and demand for, and uptake of HIV and SRH prevention and treatment technologies. The show's characters explicitly model how to discuss issues that are sensitive or taboo. MTV Shuga use the technique of 'melodramas', where drama is created through the battles between stereotypical 'goodies' and 'baddies', and the way in which the 'transitional' (often empathetic) character, begins as ambivalent but changes into a positive role model to promote positive behaviour change (15). This is a deliberate method to immerse the audience in the action, rather than passively watching or listening (13). AGYW, or at least early adopters, are anticipated to be immersed in the serial, able to classify and identify with the transitional characters and their outcomes. Pathways to behaviour change through MTV Shuga, therefore relate to the extent to which the observer, including early adopters, are immersed and critically engaged with the story. It also depends on a context which is supportive rather than disruptive (see the conceptual framework Figure 1 below) (14).

4. Last paragraph, line 13 - correct "young people" to "AGYW".

Response 4

Thank you – we have replaced 'young people' with 'AGYW' as below:

"AGYW, or at least early adopters, are anticipated to be immersed in the serial, able to classify and identify with the transitional characters and their outcomes."

Methods

1. Why are the sample sizes an approximation? Please provide the exact samples if possible.

Response methods 1

We have replaced "~2000" with 2184 and "~5000" with 4127 in the methods section – study design as follows below:

We employed a longitudinal cohort and cross-sectional surveys of representative samples of AGYW aged 12-24 in four districts of South Africa with a high burden of HIV to measure the reach of MTV Shuga-DS. Data were collected between May 2017 and September 2019.

We used baseline and follow-up data from a nested cohort of 2184 AGYW aged 13-22 years, enrolled in 2017 for the DREAMS impact evaluation. The cohort is nested in a large population-based longitudinal HIV surveillance study, in the uMkhanyakude district of KwaZulu-Natal (17, 18). A random sample of 3013 AGYW was selected from the surveillance population, stratified by age (13-17 years and 18-22 years) and geography, and invited to enrol in the nested cohort. Baseline interviews were conducted between May 2017 and February 2018 and follow-up interviews April 2018 and September 2019 in the local language (isiZulu) using a structured quantitative questionnaire programmed in REDCap (10).

The cross-sectional survey was conducted on a household-based representative sample of 4127 AGYW (between the ages 12-24 years) in three high prevalence (City of Johannesburg, Ekurhuleni and eThekweni) districts. Between August 2017 and July 2018, a stratified cluster-based sampling

strategy was used to select 18500 AGYW aged 12-24 eligible for a cross-sectional survey of individuals, based on an expected response rate of 80% (22).

2. Study design paragraph 3 - please cite where the cross-sectional survey data has already been published and/or the protocol, as you did with the nested cohort in the preceding paragraph.

Response methods 2

Thank you for this. We have now included a reference (22) for the cross-sectional protocol and included in the methods as follows:

The cross-sectional survey was conducted on a household-based representative sample of 4127 AGYW (between the ages 12-24 years) in three high prevalence (City of Johannesburg, Ekurhuleni and eThekweni) districts. Between August 2017 and July 2018, a stratified cluster-based sampling was used to select 18500 AGYW aged 12-24 eligible for a cross-sectional survey of individuals, based on an expected response rate of 80% (22).

3. Outcomes definitions lines 14-19 - This is a confusing way to state contraception methods included in the study. I recommend providing a comprehensive list of the actual methods instead.

Response methods 3

Thank you. We have provided a list of contraception methods included in the analysis as below: AGYW were considered to use contraception if they self-reported using pill, injection, Intrauterine Device (IUD), implant, sterilisation or self-reported consistent condom use (using condoms as a contraceptive method and at last sex).

4. Exposure definitions: Did you assess the number of MTV Shuga episodes watched?

Response methods 4

Thank you. We did assess the number of episodes watched but did not include it in the exposure variable because the question was only asked for those who had watched the South African series as opposed to any other MTV series in general. More than 75% of participants reported having watched it more than once.

Results

1. With over 85% of participants having not watched MTV Shuga, the exposed and not exposed groups seem unweighted to a potentially problematic degree but I don't see detail of mitigating strategies.

Response Results

We agree that our data is not balanced in terms of exposure variable, however, we used the whole random sample (cohort data with very high response rates) which is representative of the population. We also used propensity score - inverse probability weighted regression adjustment to check the robustness/consistency of our causal effect estimates. Importantly, this is a key finding a real-world evaluation. Our qualitative paper (15) provides some insight into the reasons exposure was low.

We have included the following to our statistical analysis section to further clarify:

We also used propensity score stratification and probability weighting methods to check the consistency of our findings. For cross-sectional survey, we used Chi square test to compare the characteristics of AGYW who did and did not have any exposure to MTV Shuga; and logistic regression models adjusting for potential confounders were used to examine the association between exposure to MTV Shuga and health outcomes. Sampling weights were applied to achieve proportionality between groups of participants in the survey.

Reviewer: 3

Dr. Xianhong Li, Central South University

Comments to the Author:

This study evaluate a mass-media education program in the real world settings, and the results are meaningful and have potential for further intervention designs. This manuscript is well-written. However, since the study desgin is complicated, with cohort data and cross-sectional data, the author might draw a design flow-chart to elaborate how the data are from. For example, how many data collection in the cohort? When to collect the three cross-sectional surveys data (at three time piont)? how to analysis it? It seems the manucript did not clarify this, but just use logistic regression analysis (by using which surveys?). Besides, what the rational to both using cohort data and the cross-sectional data? Please make it clear.

Response Reviewer 3

Thank you. We have revised the statistical analyses section of the paper; we separated the statistical methods for cohort and cross-sectional surveys to clarify the analyses used for the different surveys and the data collection time points. We now include a separate analysis for cross-sectional survey to clarify. The revised section now reads as follows:

Statistical analyses:

We conducted two separate analyses for cohort and cross-sectional data. For the nested cohorts, we included only participants who had data available at baseline and follow-up. We used Chi-square tests to compare baseline characteristics between AGYW who did and did not have any exposure to MTV Shuga. We used logistic regression to examine the effect of MTV Shuga on health outcomes, adjusting for exposure to DREAMS and all other potential confounders. Potential effect-modification of MTV Shuga by exposure to DREAMS was examined by fitting an interaction term to fully adjusted model: likelihood ratio tests were used to compare models with and without interaction terms.

We calculated the proportion of AGYW who reported an outcome (condomless-sex, recent pregnant) or tested positive for HSV-2 at 12-month or 24-month follow-up; and estimated associations between MTV Shuga and each outcome using a logistic regression, adjusting for potential confounders (age, household and individual socio-demographic characteristics and sexual behaviour). For HSV-2 incidence, we included participants who tested negative at baseline and had at least 1 follow-up test result. For DREAMS exposure, we included data collected at baseline and 12-month follow-up. For health outcomes (consistent condom use, modern contraception, HIV testing, PrEP awareness), we used data collected at 24-month follow-up; and for HSV-2 and pregnancy incidence we used data collected at 12 and 24-month follow-up.

Propensity score logistic regression adjustment was used to estimate the causal effect of MTV Shuga on health outcomes. A propensity or probability of being exposed to MTV Shuga was measured by fitting a logistic regression with MTV Shuga exposure as an outcome and potential confounders. A logistic regression models adjusting for propensity scores were then used to predict the probability of an outcome for all participants and separately by age group, under two scenario (1) exposed to MTV Shuga and (2) Not exposed to MTV Shuga. The predicted probabilities were then used to calculate the marginal risk difference, prevalence ratio and odds ratio. Confidence intervals were generated by using a bootstrap procedure, repeating the estimation procedure described above in 1000 samples that were drawn with replacement from the complete dataset and calculating 95% confidence intervals from the resulting bootstrap distribution using the 2.5% and 97.5% percentiles. We also used propensity score stratification and probability weighting methods to check the consistency of our findings.

For cross-sectional survey, we used Chi square test to compare the characteristics of AGYW who did and did not have any exposure to MTV Shuga; and logistic regression models adjusting for potential confounders were used to examine the association between exposure to MTV Shuga and health outcomes. Sampling weights were applied to achieve proportionality between groups of participants in the survey.

Reviewer: 1
 Competing interests of Reviewer: None
 Reviewer: 2
 Competing interests of Reviewer: I have no competing interests.
 Reviewer: 3
 Competing interests of Reviewer: no
 Editor(s)' Comments to Author (if any):

VERSION 2 – REVIEW

REVIEWER	Eaton, Andrew University of Regina, Faculty of Social Work - Saskatoon Campus
REVIEW RETURNED	09-Sep-2022

GENERAL COMMENTS	The authors have thoroughly addressed my comments and the manuscript is much improved.
--

REVIEWER	Li, Xianhong Central South University, Xiangya School of Nursing
REVIEW RETURNED	29-Sep-2022

GENERAL COMMENTS	Authors have thoroughly revised the manuscript, and addressed most of the comments. I think it is ready for publication, except some couple more concerns below:  1.The bulletin # 3 seems not the strength of the study. 2.The bulletin # 4 is not an intact sentence, and the logic is not clear: “in this case exposure to MTV-Shuga and impact on uptake of sexual health promotion and innovative technologies”. 3.As for the Review 2’s comment: “Paragraph 2 - Implied assumption that all AGYW are heterosexual and sexually active is problematic”, I think the authors did not addressed it enough. The comments states that assuming all the AGYW are heterosexual and sexually active is problematic, which means that homosexual AGYW is neglected. I think authors can add this as a limitation.
--

VERSION 2 – AUTHOR RESPONSE

Outstanding Reviewer 3 comments

Reviewer: 3

Dr. Xianhong Li, Central South University

Comments to the Author:

Authors have thoroughly revised the manuscript, and addressed most of the comments. I think it is ready for publication, except some couple more concerns below:

Reviewer comment 1

1.The bulletin # 3 seems not the strength of the study.

Response 1

We have rephrased bullet #3 to show how it is a strength of the study. Currently the bullet reads “Study measured the reach of MTV-Shuga and the relationship between exposure to the edu-drama on awareness and demand for HIV prevention technologies in representative longitudinal cohort and cross-sectional samples of young women.”

Now it reads:

- The strength of this study is the use of longitudinal data from a cohort of AGYW nested in a larger population-based longitudinal HIV surveillance. In addition data were drawn from cross-sectional representative samples of AGYW from four districts of South Africa. These data enabled us to measure the reach of MTV-Shuga and the relationship between exposure to the edu-drama on awareness and demand for HIV prevention technologies in young women.

Reviewer comment 2

2.The bulletin # 4 is not an intact sentence, and the logic is not clear: “in this case exposure to MTV-Shuga and impact on uptake of sexual health promotion and innovative technologies”.

Response 2

Thank you for pointing this out. We have revised bullet #4. The initial one read “Observational studies give no opportunity to infer the cause–effect relationship, in this case exposure to MTV-Shuga and impact on uptake of sexual health promotion and innovative technologies.”

Now it reads:

- The limitation of observational studies is that they do not infer the cause–effect relationship, in this case we cannot ascertain causality/impact of exposure to MTV-Shuga on uptake of sexual health promotion and innovative technologies.

Reviewer comment 3

3.As for the Review 2’s comment: “Paragraph 2 – Implied assumption that all AGYW are heterosexual and sexually active is problematic”, I think the authors did not address it enough. The comments states that assuming all the AGYW are heterosexual and sexually active is problematic, which means that homosexual AGYW is neglected. I think authors can add this as a limitation.

Response 3

Thank you for raising this concern and again we acknowledge this was a shortcoming of the DREAMS approach. However, in this study we were evaluating the effect of MTV-Shuga on young women’s uptake of SRH services in the context of DREAMS roll-out. MTV-Shuga targeted both

heterosexual and homosexual young people and had a storyline on sexual identity which included a gay character. We have included sexual identity as a component of MTV-Shuga in our description of MTV-Shuga to make it clearer [page 4, last paragraph,page 5, first sentence]. It now reads:

“From March 8th 2017, MTV Shuga aired one episode per week for 12 weeks (with repeats). MTV Shuga is a mass media behaviour change campaign that aims to improve sexual and reproductive health rights (SRHR). At the centre of the campaign, which includes radio and social media, is a TV-drama that weaves messages about HIV, family planning, transactional and intergenerational sex, sexual identity, safer and healthy sexual relationships, into storylines with young characters (http://www.mtvshuga.com/show/series-5/MTV_Shuga-down-south/).”

Reviewer 1 Reviewer: Sarah Hawi Ngere, BA, MPH

Review summary

According to my understanding the study was set to investigate the effect of exposure to MTV Shuga Down South (DS) during the scale-up of combination HIV-prevention interventions (DREAMS) on awareness and uptake of SRH and HI prevention services by adolescent girls and young women (AGYW). The data was collected through a defined period of time. The researchers have concrete understanding of the topic area. I have only minor comments the manuscript is well written with clarity on all aspects.

Section by section comments

Page 2 Abstract Line 11: Under subsection, objectives, DREAMS acronym appears for the first time but authors do not give the meaning of what acronym stands for.

Comment: No further comments. This section is well written.

Response

Thank you. We have edited the abstract to remove the need for this Acronym.

Page 4-6 Introduction The authors did a good job in this section, introducing the readers to the background of the problem and systematically stating out the problem and what gaps the study intends to fill

Response

Thank you.

Methods General comment/observation: I find this section well written. The design has rigor and reliability.

Response

Thank you.

Page 10 line 52-60 and page 11 line 1-13: Ethics, Ethical consideration for data management is missing.

Suggestion: Consider including a sentence to describe how the data collection and storage were ethically sound.

Response

We have included a sentence on page 20 last paragraph to indicate that participation was voluntary, and consent was obtained prior to surveys, and data was anonymized for analysis, it now reads:

“Participation in the study was voluntary. All participants provided separate informed consent to take part in data collection through the questionnaires and the HSV2 sero-survey. Consent for follow-up interviews was provided separately. For participants aged <18 years, written parental consent and participant assent were provided. All data were anonymised for analysis.”

Results

General comment/observation: Other than one comment stated below the section is clearly written.

Awareness and exposure to MTV Shuga, “*MTV shuga exposure at baseline was low*” Since exposure is not clearly defined on page 8 line 34 -52 I found it difficult to understand “low exposure” as stated in the line above because of how it was described therefore I think it should be either exposed or not exposed to MTV Shuga (please note that I understood there was a difference between exposure and level of exposure).

Response

We have added the suggestion to the sentence, now it reads:

MTV Shuga exposure at baseline was limited, with a total of only 308 (14.1%) respondents reported watching at least one episode.

Page 17 Discussion General comment/observation: The discussion section is well written with all major findings adequately discussed.

Response

Thank you.

Page 18 Limitations General comment/observation: The limitation section indicates enough evidence to guide future studies on the gaps that the study was unable to fill. I find the section well written.

Response

Thank you.

Line 51 "...effect we did not have the power to show significant..." this sentence may be missing "**but**" therefore the statement can be state as: "...effect **but** we did not have the power to show significant..."

Response

Thank you we have revised and now it reads:

"...suggested a possible effect but we did not have the power to show a significant relationship between exposure to MTV Shuga and SRHR outcomes."

Conclusion and implications for the future General comment/observation: No comments for this section.

Response

Thank you.